Land use/land cover changes in the central part of the Chitwan Annapurna Landscape, Nepal

Adhikari Jagan Nath 1 2
http://orcid.org/0000-0001-5741-6179 Bhattarai Bishnu Prasad 1 bhattaraibp@gmail.com
Rokaya Maan Bahadur 3 4
Thapa Tej Bahadur 1 tej.thapa@cdz.tu.edu.np
1 Central Department of Zoology, Institute of Science and Technology, Tribhuvan University Kathmandu , Kirtipur, Bagmati , Nepal
2 Department of Zoology, Birendra Multiple Campus, Tribhuvan University, Chitwan , Bharatpur, Bagmati , Nepal
3 Global Change Research Institute, Czech Academy of Sciences , Brno, Moravia , Czech Republic
4 Institute of Botany, Czech Academy of Sciences , Průhonice , Czech Republic
Provete Diogo
Electronic publication date: 2022 May 20
Publication date: 2022
Volume: 10
Electronic Location ID: e13435
Received 2021 Dec 20; Accepted 2022 Apr 22
Copyright: © 2022 Adhikari et al.
Copyright year: 2022
Copyright holder: Adhikari et al.
License: This is an open access article distributed under the terms of the Creative Commons Attribution License, which permits unrestricted use, distribution, reproduction and adaptation in any medium and for any purpose provided that it is properly attributed. For attribution, the original author(s), title, publication source (PeerJ) and either DOI or URL of the article must be cited.
License URL: https://creativecommons.org/licenses/by/4.0/

Keywords: Accuracy assessment, Habitat change detection, Image classification, Landsat image, Remote sensing

Funding: The authors received no funding for this work.

==============================
Background

Land use/land cover assessment and monitoring of the land cover dynamics are essential to know the ecological, physical and anthropogenic processes in the landscape. Previous studies have indicated changes in the landscape of mid-hills of Nepal in the past few decades. But there is a lack of study in the Chitwan Annapurna Landscape; hence, this study was carried out to fill in study gap that existed in the area.

Methods

This study evaluates land use/land cover dynamics between 2000 to 2020 in the central part of the Chitwan Annapurna Landscape, Nepal by using Landsat images. The Landsat images were classified into eight different classes using remote sensing and geographic information system (GIS). The accuracy assessment of classified images was evaluated by calculating actual accuracy, producer’s accuracy, user’s accuracy and kappa coefficient based on the ground-truthing points for 2020 and Google Earth and topographic maps for images of 2010 and 2000.

Results

The results of land use/land cover analysis of Landsat image 2020 showed that the study area was composed of grassland (1.73%), barren area (1.76%), riverine forest (1.93%), water body (1.97%), developed area (4.13%), Sal dominated forest (15.4%), cropland (28.13%) and mixed forest (44.95%). The results of land cover change between 2000 to 2020 indicated an overall increase in Sal dominated forest (7.6%), developed area (31.34%), mixed forest (37.46%) and decrease in riverine forest (11.29%), barren area (20.03%), croplands (29.87%) and grasslands (49.71%). The classification of the images of 2000, 2010 and 2020 had 81%, 81.6% and 84.77% overall accuracy, respectively. This finding can be used as a baseline information for the development of a proper management plan to protect wildlife habitats and forecasting possible future changes, if needed.

Introduction

Land use/land cover changes (LULCC) are widely evaluated in different parts of the world as a result of increasing socio-economic necessities needed for ever increasing human population (Hassan et al., 2016; Reis, 2008; Zhu et al., 2021). The LULCC leads to change in vegetation cover and other different components of biodiversity (Halmy, Fawzy & Nasr, 2020; Petrou, Manakos & Stathaki, 2015). It is, thus, important to know the extent of LULCC to find out the drivers and their exact impacts on ecological (e.g., forest cover) and anthropogenic processes (e.g., cropland and settlement area). LULCC are the major sources of environmental changes such as change in biodiversity, habitats, destructions, loss of soil resources, landslides, flood, global climate change and the impact of invasive and alien plant species (MEA, 2005; Nepal Ministry of Land Reform and Management (MoLRM), 2015; Paudyal et al., 2019; Rather, Kumar & Khan, 2020; Rimal et al., 2019; Wu, 2019). Hence, understanding about LULCC is important issues in current scenario (Chamling & Bera, 2020).

The landscape is spatially heterogeneous and composed of the visible features of a geographic area (Crowley & Cardille, 2020; Shao & Wu, 2008) that is directly or indirectly affected by ecological (e.g., biotic interactions, ecological successions), physical (e.g., natural disasters) and anthropogenic (e.g., agricultural practices, livestock grazing) factors (Rather, Kumar & Khan, 2020; Scheller, 2020; Zhu et al., 2021). Land use relates to land cover patterns, and it affects to numerous consequences (Siddique et al., 2020). Landscape patterns quantify the configuration and composition of the landscape by using the number of matrices which are further used for the distribution of the species (Haines-Young, 1992; Raut, Chaudhary & Thapa, 2020). The studies revealed that anthropogenic factors cause more change in land cover use than the environmental factors (Rimal et al., 2019; Song et al., 2018).

The mid-hills of Nepal are human dominated and highly fragmented. In the past, the people lived in rural area and performed agricultural activities. They cleared the forest for the expansion of agriculture, hence, the forest was in decreasing trend (Nepal Ministry of Forests and Environment (MoFE), 2019) but now, the scenario has been changed. About one-third of agricultural land in the mid-hills of Nepal has already been abandoned and the people migrate to the urban and semi-urban areas (Garrard et al., 2016; Paudel, Dahal & Shah, 2012). This migration process leads to increase the forest cover in the rural area and population growth, unplanned expansion of settlements, increased demand of natural resources in urban and semi-urban areas. The policy makers seek the information on the causes and main effects of LULCC for developing the policies as well as a management plan for the conservation of natural resources.

Studies related to the LULCC in Nepal have focused mainly on the urbanization patterns (Thapa & Murayama, 2009; Wang et al., 2020), glacier fluctuations and outburst, and landslides (Huggel et al., 2002; Rimal et al., 2019; Sharma et al., 2019), land cover change in and around the watershed and river systems (Lamsal et al., 2019; Paudyal et al., 2019; Rai et al., 2018) and land use/land cover change in the protected areas (Chettri et al., 2013; Kafley, Khadka & Sharma, 2009; Thapa, 2011). However, there are scattered information on the studies at landscape level and the studies related to land cover change analysis are not adequate in number (Chhetri, Shrestha & Cairns, 2017; WWF, 2013a; Zomer, Ustin & Carpenter, 2001). LULCC data sets provide detailed information about ecosystems and processes needed for analysis and modeling (Rather, Kumar & Khan, 2020; Rimal et al., 2019; Wang et al., 2020). Hence, this study classified the temporal and spatial pattern of LULCC in the central part of the Chitwan Annapurna Landscape (CHAL), Nepal.

Materials and Methods

Study area

The CHAL in the central Nepal is drained by eight major rivers (Kali Gandaki, Seti, Madi, Marshyandi, Daraudi, Budi Gandaki, Trishuli and Rapti) and their tributaries. This landscape covers all or parts of six protected areas and 19 districts (WWF, 2013a). We have chosen the central part of CHAL that connects two biologically important protected areas, the Chitwan National Park (CNP) in the south and the Annapurna Conservation Area (ACA) in the north. This part of CHAL has given the highest priority corridor for landscape level connectivity (WWF, 2013b). The intensive study area covers Chitwan (around Barandabhar Corridor and surrounding areas), Tanahun (Seti River basin), Kaski and some parts of Syanja and Parbat districts (Panchase and part of Annapurna Conservation Area) with an area of 2,749.48 km2 (Fig. 1). The elevation ranges from 150 to 3,300 m. The lowland part has tropical and subtropical types of climate, whereas mid-hills have the temperate type of climate and the upper mountain region has subalpine type has of climate.

Figure 1 Map showing the intensive study areas which links two biodiversity significant areas: Chitwan National Park (CNP) and Annapurna Conservation Area.

This landscape is rich in biodiversity, including three Global 846 Ecoregions (Terai–duar Savanna and Grasslands, Himalayan Subtropical Broadleaf Forests, Himalayan Sub-tropical pine forest) (Dinerstein et al., 2017; Wikramanayake, Dinerstein & Loucks, 2002) and two Ramsar sites (Beeshazari and associated lakes, Chitwan and Lake Clusters of Pokhara valley, Kaski) (National Lake Conservation Development Committee of Nepal (NLCDC), 2020). This area is prime habitat for many important mammal species, birds, herpetofauna, fish and many other micros and macroinvertebrates (Bhuju et al., 2007; WWF, 2013b).

Data sources

Landsat images from 2000, 2010 and 2020 were used to detect the LULCC within the 10-year time interval. The Landsat 7-ETM (Enhanced Thematic Mapper) for 2000, Landsat 5-TM (Thematic Mapper) for 2010, and Landsat 8-OLI (Operational Land Imager) for 2020, images with same 30 m spatial resolution were downloaded from the United States Geological Survey (USGS) (https://glovis.usgs.gov/app) geoportal. A total of six scenes of satellite images of two from each year were downloaded (Table 1). The entire Landsat images consist of around 3–10% of cloud cover, but this was less than 1% in our study area. We also used the topographic maps with 1:25,000 and 1:50,000 scales developed by the Department of Survey, Government of Nepal.

Table 1 List of dataset used in the study.

SN	Acquisition date	Data category	Spatial resolution	Band properties	Sources	
1	3 April 2000	Landsat 7, Enhanced Thematic Mapper (ETM)	30 m	Multispectral	https://glovis.usgs.gov/app	
2	18 February 2010	Landsat 5, Thematic Mapper (TM)	30 m	Multispectral	https://glovis.usgs.gov/app	
3	17 March 2020	Landsat 8, Operational Land Imager (OLI)	30 m	Multispectral	https://glovis.usgs.gov/app	
4	1999/2000	Topographic map	1:25,000
1:50,000		Department of survey, Kathmandu	
5	2018–2020	Ground truth (reference data)			Field survey- GPS	
6	2000, 2010, 2020	Google Earth Pro			https://earth.google.com/web/	
7	2010	ICIMOD		Classified	http://rds.icimod.org/	

In addition, the Google Earth and a classified map of 2010 developed by the International Centre for Integrated Mountain Development (ICIMOD) (http://rds.icimod.org) used as a reference for verification. The reference field data were collected using a Global Positioning System (GPS) during the field study and used as ground-truthing points during classification of images and accuracy assessments (Table 1).

Image pre-processing

Each band of Landsat image was checked using metadata and georeferenced to the WGS_84 datum and Universal Transverse Mercator (UTM) Zone 44 or 45 North coordinate system. The details of bands and resolution are mentioned in Table S1. We only used the bands with 30 m resolution for further analysis. Landsat 5 TM images have seven bands, Landsat 7 ETM images have eight spectral bands, and Landsat ETM has 8 bands in which 1 to 7 bands have 30 m resolution (Barsi et al., 2014). Similarly, Landsat 8 OLI images have 11 bands in which eight bands 1 to 7 and 9 have 30 m resolution. (https://www.usgs.gov) (Table S1). For the natural color composite of Landsat 8 OLI images, band 4 (red), 3 (green) and blue (2) were combined for the natural color, whereas bands 7, 6, 4 were used for false color (urban). Similarly, bands 5, 4, 3 for vegetation composition, bands 6, 5, 2 for agriculture, 5, 6, 4 for land and water (Vermote et al., 2016; Barsi et al., 2014).

The images were processed in ERDAS IMAGINE 9.2. Bands of each satellite image (2000, 2010 and 2020) were stacked within Raster main icon with layer stack function as a single layer. In this study, we selected band 1 to 5 and 7 (blue, green, red, near infrared (NIR), shortwave infrared I (SWIR1) and shortwave infrared II (SWIR2)) for Landsat 5 TM and Landsat 7 ETM; band 1 to 7 (coastal, blue, green, red, NIR, SWIR1 and SWIR2) for Landsat 8 OLI in land use and land cover classification. Band 8 to 11 of Landsat image 2020 are less used in LULCC (Yu et al., 2019). The images of each scene were masked using the Area of Interest (AOI) of the study area using mask function (Fig. 2).

Figure 2 Flow chart of overall process of Landsat image classification.

Ground-truthing points

For the data collection (ground truthing coordinates) in the central part of Chitwan-Annapurna Landscape, we obtained permission from the Department of National Parks and Wildlife Conservation (Permission letter number 3372), Chitwan National Park (Permission letter number 2723), Division Forest Offices of Chitwan (Permission letter number 2723), Tanahun (Permission letter number 749), Kaski (Permission letter number 200) districts and Annapurna Conservation Area Project (Permission letter number 66). The field survey which was carried out from 2018 to 2020 provided a clear idea about the field, forest types and land cover types. For ground-truthing, geographic coordinates were collected during the sign survey of large mammals, including leopard and their prey using GPS (Garmin eTrex 10). These geographic coordinates represented all land cover types along the landscape. Each coordinate was taken from the central point of the land cover patches which was more than 30 m radius. A total of 1,350 coordinates were collected (259 from Sal dominated forest, 125 from riverine forest, 299 from mixed forest, 125 from grasslands, 88 from barren areas, 135 from developed areas, 92 from water bodies and 229 from cropland). Out of the total sampling coordinates, half of the coordinates (667) were used for supervised classification and the remaining coordinates (683) were used for accuracy assessment. In addition to this, we also used printed versions of topographic maps to locate the different land cover types including changes over there through participatory GIS (pGIS) techniques. pGIS studies consider that the local people are familiar and experience with change to their surroundings and provide the greater spatial information about the area (Aynekulu et al., 2006; Zolkafli, Brown & Liu, 2017). For this purpose, focus group discussions were performed with members of community forests and elderly people who inhabited for a long time in that area and easily felt the changes in their surroundings. Twenty group discussions were arranged in different locations of the landscape (five discussions on Barandabhar and associated area, ten on the Seti River basin of Tanahun, five on Panchase and lower part of the ACA).

Image classification

The consistency of the land cover classes at national, regional and international level is not same (Chettri et al., 2013; Nepal Ministry of Land Reform and Management (MoLRM), 2015; Uddin et al., 2015b; Wang et al., 2020). In the present study, land cover classification was established with the help of published literature and maps (Khanal et al., 2020; Nepal Ministry of Forests and Environment (MoFE), 2019; Nepal Ministry of Land Reform and Management (MoLRM), 2015; Thapa, 2011; Uddin et al., 2015b; Zomer, Ustin & Carpenter, 2001). We classified the land cover of the central part of CHAL into eight major classes, based on the dominant plant species, human settlements, landscape and agriculture. We categorized the forest types as Sal dominated forest, riverine forest and mixed forest (Table 2). The dominant plant species composition in the mid-hills is of mixed type and difficult to separate into other subcategory, hence, we classified such forest as mixed forest.

Table 2 Major land use and land cover types in the central part of the Chitwan Annapurna Landscape, Nepal.

SN	Land cover types	Description	
1	Water bodies	River, lakes, ponds, marshy land	
2	Barren area	Sand, gravel, dry beds, flood plains without vegetation, landslide, snow feed area and no vegetation areas	
3	Grassland	Grasslands, scattered shrub	
4	Riverine forest	Simal (Bombax ceiba), Khair (Acacia catechu), Sisso (Dalbergia sissoo), Veller (Trewia nudiflora), Padke (Litsea doshia), Kutmero (Litsea monopetala) and associates plants	
5	Sal dominated forest	Sal (Shorea robusta), Saj (Terminalia alata), Karma (Adina cordifolia) and associates plants	
6	Mixed forest	Dhairo (Woodfordia fruticosa), Kyamuno (Syzygium cumini), Amaro (Spondias pinnata), Chilaune (Schima wallichii), Katus (Castanopsis tribuloides), Kafal (Myrica esculenta), Utis (Alnus nepalensis), Paiyu (Prunus cerasoides), Ritha (Sapindus mukorossi), Lapsi (Choerospondias axillaris), Champ (Michelia champaca), Rakchan (Daphniphyllum himalayense), Rhododendron and oak (Quercus spp), and associate plants	
7	Cropland	Crop (e.g. paddy, maize, millet, mustard, wheat etc.) cultivated lands	
8	Developed area	Urban and rural settlements, commercial areas, industrial areas, hydropower project areas, roads construction, airport	

Unsupervised classification

In the beginning, the unsupervised classification of the multi-temporal Landsat images of 2000, 2010, 2020 was performed. This classification is based on the automatic identification and assignment of image pixels to spectral grouping. It starts with a spectral plot of the whole image and group the pixels with similar features. Two common algorithms are used for the creation of the clusters in unsupervised classification (Duda & Canty, 2002). They are k-means clustering and Iterative Self-Organizing Data Analysis Technique (ISODATA) (Ragettli, Herberz & Siegfried, 2018). In this classification, we used k-means algorithm. The nearest likelihood with 10 iterations were used to group the pixels having similar features. The images were classified into 40 classes with a convergence threshold 0.90. Then, the similar classes were merged into eight different classes using recoding of classes (Table 2 and Fig. 2). The unsupervised classification of images was used for the planning of field data collection that provided the basic field knowledge. The unsupervised classes were revised after the collection of ground-truthing points.

Supervised classification

The supervised classification was performed using the widely used parametric classification algorithm namely Maximum Likelihood Classification (MLC) (Chamling & Bera, 2020; Rai et al., 2018). The signature classes or training sets were prepared from ground-truthing points for 2020 and Google Earth map for 2000 and 2010 were used to prepare signature classes for supervised classification. Two separately classified Landsat images were mosaicked to make a single image. Finally, the images were filtered fixing the pixels 3 × 3 for smoothing the image and avoid the errors of misclassification. The images were again recoded based on field knowledge to minimize the errors of misclassification. We selected five sites, two from low land (Barandabhar and associate area), two from mid hill (Seti River basin and Panchase area) and one from an upland area (lower part of the ACA) for the separate analysis where the land use/land cover was changed drastically within the land 20 years.

Accuracy assessment

Accuracy assessment increases the quality of the remotely sensed data on classified thematic maps. It compares the classified image with ground truthing points (Congalton, 2001; Rai et al., 2018; Siddique et al., 2020; Song et al., 2001; Thapa, 2011). Another common method to assess the accuracy of the classified map is to generate stratified random points as the classified class. These random points compared with the Google Earth and topographic maps as reference for verification (Crowley & Cardille, 2020). The topographic maps of Nepal were used as reference of settlements or developed area, water resources, croplands and forest area. In this study, ground-truthing points (n = 683) were used as reference for the accuracy assessment of classified images of 2020 (Fig. 3). For Landsat images of 2000 and 2010, 500 stratified random points were generated and compared them with references such as Google Earth, topographic maps of Nepal (for water bodies, settlements, urban or developed area and forest) and the classified maps of ICIMOD (for the classification of forest and grassland). The evaluation was performed computing confusion matrix or error matrix and Kappa Coefficient (Congalton, 2001; Foody, 2002). The user’s accuracy, producer’s accuracy, overall accuracy was obtained from the error matrix. The user’s accuracy provides the reliability that the classified pixels of the map match with the ground-truthing points (Eq. (2)). Similarly, the producer’s accuracy determines the probability of correctly classified reference pixels (Eq. (3)). The overall accuracy was calculated by dividing the correctly classified pixels by the total number of reference points (Eq. (1)) (Congalton, 2001; Foody, 2002). Kappa Coefficient ( K^) is used to measure the agreements between model prediction and reality (Congalton, 2001). It is the multivariate analysis technique to evaluate the accuracy of the classified map statistically. The Kappa Coefficient ( K^) ranges from 0 to 1. If the value of K^ is 0, this reflects there is no agreements, 0–0.2 signifies as slight, 0.21–0.40 as fair, 0.41–0.60 as moderate, 0.61–0.80 as satisfactory or good and 0.81 to 1 as almost perfect agreements (Maingi, Kepner & Edmonds, 2002). Statistically, the K^ was calculated using Eq. (4).

Figure 3 Map showing the ground-truthing points used for accuracy assessment of the classified land cover image of 2020.

(1) Overallaccuracy=TotalnumberofcorrectlyclassifiedpixelsTotalnumberofreferencepixels×100

(2) User′saccuracy=NumberofcorrectlyclassifiedpixelsineachcategoryTotalnumberofclassifiedpixelsthatcategory(rowtotal)×100

(3) Produceraccuracy=NumberofcorrectlyclassifiedpixelsineachcategoryTotalnumberofclassifiedpixelsthatcategory(columntotal)×100

(4) Kappacoefficient(K^)=N(∑i=1r⁡Xii)−∑i=1r⁡(X1+×X+i)N2−∑i=1r⁡(Xi+×X+i)

where, r = Number of rows in the error matrix

Xii = number of observations in row i and column i (on the major diagonals)

Xi+ = Total number of observations in rows i

X+i = Total number of observations in column i

N = Total number of observations included in matrix

Results

Land use/land cover classes and change

Out of eight land cover classes of 2020, mixed forest was the most dominant (44.95%) followed by croplands (28.3%), Sal dominated forest (15.4%) and developed area (4.13%) (Table 3, Fig. 4).

Table 3 Land cover classes in the central part of Chitwan-Annapurna Landscape in 2020.

SN	Land cover type	Area_2020 (Km2)	Percentage	
1	Water bodies	54.04	1.97	
2	Barren area	48.62	1.76	
3	Grassland	47.32	1.73	
4	Riverine forest	53.25	1.93	
5	Sal dominated forest	423.65	15.4	
6	Mixed forest	1235.9	44.95	
7	Cropland	753.35	28.13	
8	Developed area	113.35	4.13	
	Total area	2749.48	100	

Figure 4 Land cover types of the central part of the Chitwan-Annapurana Landscape in 2020.

The results of LULCC from 2000 to 2010 indicated that there was a decrease in water bodies, barren land, grassland, riverine forest and croplands by 0.9%, 7.7%, 6.2%, 13% and 16%, respectively; build-up or developed area, Sal dominated forest and the mixed forest were increased by 19.1%, 4.62% and 18.2%, respectively. Similarly, from 2010 to 2020, water bodies, riverine forest, Sal dominated forest, developed area and mixed forest were increased by 2.54%, 2.09%, 3%, 10.3% and 16.3%, respectively. Barren area, cropland and grasslands were decreased by 13.3%, 16.3% and 46.4% respectively (Table 4, Figs. 5 and 6). Overall, from 2000 to 2020, the areas of grassland, riverine forest, cropland and barren area were drastically decreased, whereas developed area, mixed forest and Sal dominated forest were increased (Table 4).

Table 4 Land cover changes in study area from 2000 to 2020.

SN	Land cover type	Land cover area (km2)	Change 2000–2010	Change 2010–2020	Change 2000–2020	
		2000	2010	2020	Area	%	Area	%	Area	%	
1	Water bodies	53.2	52.7	54.04	−0.5	−0.9	1.34	2.54	0.84	1.57	
2	Barren area	60.8	56.1	48.62	−4.7	−7.7	−7.48	−13.3	−12.2	−20.03	
3	Grassland	94.1	88.24	47.32	−5.86	−6.2	−40.9	−46.4	−46.8	−49.71	
4	Riverine forest	60.03	52.16	53.25	−7.87	−13	1.09	2.09	−6.78	−11.29	
5	Sal dominated forest	393.15	411.3	423.65	18.15	4.62	12.4	3	30.5	7.76	
6	Mixed forest	899.1	1062.48	1235.9	163.38	18.2	173	16.3	337	37.46	
7	Cropland	1102.8	923.7	773.35	−179.1	−16	−150	−16.3	−329	−29.87	
8	Developed area	86.3	102.8	113.35	16.5	19.1	10.6	10.3	27.1	31.34	
	Total	2749.48	2749.48	2749.48							

Figure 5 The land-use/cover change in area during the period of 2000–2020.

Figure 6 Land cover change between (A) 2000, (B) 2010 and (C) 2020.

The separate analysis of LULCC between 2000 to 2020 in old Padampur and associated areas (low land) clearly showed that more than 93% of the total cultivated land was changed into the grassland and forest. Similarly, the barren area (flood plain of Rapti River) was reduced by 74.67%. However, grassland, riverine forest and mixed forest in the old Padampur and associated areas were increased by 94.45%, 91.26% and 62.5%, respectively (Figs. 7A1–7A3, 8A, Table S2). The trend of land cover change from 2000 to 2020 in new Padampur and associated areas (low land) indicated that the riverine forest, Sal dominated forest and grassland were drastically reduced by 61.21%, 54.14% and 64.88%, respectively, whereas the cropland and developed areas were increased by 88.17% and 1433.33%, respectively (Figs. 7B1–7B3, 8B, Table S2). Land cover change from 2000 to 2020 in Byas municipality of Tanahun district and surrounding areas showed a significant reduction in the cropland by 40.86%, whereas there was a significant increase in developed areas and mixed forest by 86.55% and 62.14%, respectively. The trend of land cover change in Byas and surrounding areas was more between 2010 to 2020 than 2000 to 2010 (Figs. 7C1–7C3, 8C, Table S2). The results of land cover change analysis of Panchase Protected Forest and associate areas between 2000 to 2020 showed a reduction in cropland by 51.92% and grassland by 43.22%, whereas an increase in mixed forest and Sal dominated forest by 68.1% and 23.29%, respectively (Figs. 7D1–7D3, 8D, Table S2).

Figure 7 Synergic change in land cover in the part of study area from 2000 to 2010. Here, (A1–A3) land cover change in Old Padampur area; (B1–B3) land cover change in New Padampur area; (C1–C3) land cover change in Byas area; (D1–D3) land cover change in Panchase Protected Forest and associated area; (E1-E3) land cover change in the part of Annapurna Conservation Area.

Figure 8 Percentage of Land cover change from 2000 to 2020. Here, (A) Old Padampur area; (B) New Padampur and associated area, (C) Byas and associated area (an example of populated area of mid-hill); (D) Panchase and associated areas (an example of rural area of Midhill), (E) a part of Annapurna Conservation Area (an example of rural area of Mountain, inside the protected area).

The results of land cover change analysis of a part of the ACA between 2000 to 2020 clearly showed an increase in mixed forest and developed area by 14.93% and 166.66%, respectively, whereas a decrease in cropland, barren area and grassland were decreased by 40.97%, 24.09% and 19.94%, respectively (Figs. 7E1–7E3, 8E, Table S2).

Accuracy assessment

The overall accuracy of classified images of 2000, 2010 and 2020 was 81%, 81.6% and 84.77%, respectively. The user’s accuracy ranged from 73.33% to 87.09% in 2000, 73.68% to 83.33% in 2010 and 80.26% to 90.69% in 2020. The low range of user’s accuracy in barren area in 2000 (73.33%), in a developed area in 2010 (73.68%) indicated confusion during land cover classification (Tables 5). Riverine forest in 2000, mixed forest in 2010 and Sal dominated forest in 2020 were more reliable with user accuracy of 87.09%, 83.77% and 90.69%, respectively (Tables 5, Tables S3–S5). The Kappa coefficient for the years 2000, 2010 and 2020 were 0.76, 0.79 and 0.82, respectively.

Table 5 Accuracy assessment of the classified images from 2000–2020.

Land cover	2000	2010	2020	
User’s accuracy	Producer’s accuracy	User’s accuracy	Producer’s accuracy	User’s accuracy	Producer’s accuracy	
Water bodies	81.81	90	76.92	76.92	90	81.18	
Barren area	73.33	73.33	80	72.73	82	69.49	
Grass land	78.37	80.5	75	80	80.95	76.11	
Riverine forest	87.09	81.8	76.92	71.4	84.61	84.61	
Sal dominated forest	84.21	80	83.11	80	90.69	95.9	
Crop land	82.73	80.41	83.33	83.3	85.32	83.78	
Developed area	77.77	72.41	73.68	66.67	84.62	80.88	
Mixed forest	78.43	83.3	83.77	86.95	80.26	89.7	
Over all accuracy	81	81.6	84.77	
Kappa coefficient	0.76	0.79	0.82	

Discussion

The present study categorized eight land cover classes including four major forest types- Sal dominated forest, riverine forest, mixed forest and grassland. Among the land cover classes, Sal dominated forest was the most common in the Barandabhar Corridor Forest and some parts of Tanahun and Kaski districts. The tropical and subtropical climate with high temperature and precipitation support the Sal dominated forest (Adhikari, Bhattarai & Thapa, 2019; Reddy et al., 2018). Similarly, the riverine forest was found in the flood plains of major river systems (Rapti, Narayani, Marshyandi, Kaligandaki, Seti river basin). In the mid-hills, most of the area was covered by mixed forest. LULCC analysis in the central part of the Chitwan Annapurna Landscape showed that there were more than 62% of total land covered by forest area (mixed forest, Sal dominated forest and riverine forest). Therefore, this area is regarded as priority corridor for biodiversity conservation in CHAL. However, this landscape is human-dominated and highly fragmented (WWF, 2013a) due to the scattered human settlements and croplands. The river systems (Rapti, Narayani, Seti, Madi, Modi, Kaligandaki, Marshyandi and other associates) and lakes (two Ramsar sites Beeshhazari Lake and Lake clusters of Pokhara Valley) are crucial for maintaining different ecosystems. Similar type of study based on the Google Earth map analysis of 2018 by Nepal Ministry of Forests and Environment (MoFE) (2019) found 44.47% of the total area in Nepal was covered by forest.

The temporal patterns of the LULCC analysis showed the direction of land cover changes with respect to the initial land cover (land cover of 2000) as a reference. Our classified images of the central part of CHAL clearly showed a decrease in cropland (29.87%) and drastically an increase in mixed forest (37.46%). This is due to the shifting of the people from the hilly area to the urban area for a better life and employment opportunities, hence, the cropland left by them gradually converted into the forest (Garrard et al., 2016). Such type of changes was observed in the studies; Bhandari et al. (2022) in Bhanu Municipality, Ragettli, Herberz & Siegfried (2018) in Tanahun district and Kc & Race (2019) in Lamjung district of Nepal. The results of increment of urban area from 2000 to 2020 (31.34% increment) also proved the migration of the people from rural to urban area as the study by Kc & Race (2019). The rapid development of the roads, tracks, hydropower, industrial areas, airports and settlements in urban areas have created major barriers for wildlife movements. The settlement density was more in urban and plain areas than in the hilly areas (CBS, 2012). Similarly, the study of Ragettli, Herberz & Siegfried (2018) indicated the increase in the barren area in Tanahun district between 2000 to 2019 but our study indicated the decrease in the barren and grassland area within the landscape because the most of these areas were replaced by the forest. The grasslands that were scattered inside the forest and the grassland in the mountain were used by the local people as pasture land as reported in the study of Rai et al. (2018) in Gandaki River basin and Chetri & Gurung (2004) in Upper Mustang in the central Nepal. The landslide was very common in the mid-hills and high mountain (Budha et al., 2020; Petley et al., 2007). Besides, rivers also deposited sands and gravels to their catchment areas, played a significant role in land cover change.

The increase in the forest indicated that there are improvements in wildlife habitats, especially for large mammals. Forest cover inside the protected areas (Chitwan National Park and Annapurna Conservation Area) was also in increasing steadily as observed in the Old Padampur area. After the shifting of Padampur village to another place to include an old village area inside the Chitwan National Park, the crop land was transformed into the grassland and riverine forest. The land cover change analysis showed that more than 94% grassland was increased from 2000 to 2020 in the Old Padampur area. The forest was cleared and the Padampur village was relocated to the New Padampur area. Hence, the cropland and developed areas increased drastically within the period of 20 years in newly settled areas. Similarly, the forest increased in the mid-hills due to the implementation of effective community forestry program by the government. Our findings were similar to the findings of other parts of Nepal such as in Nepal’s Kailash Scared Landscape (Uddin et al., 2015a), Koshi River basin (Rimal et al., 2019) and Mechinagar and Buddhasanti landscape (Rijal et al., 2021b) but different than studies from Bagmati River basin (Rijal et al., 2021a). Regeneration of the forest inside the ACA increased during recent years. The people abandoned the marginal agriculture land due to low production, shortage of labors for agricultural work and high human wildlife conflict, hence, these areas were converted into the forests. Similar observations were found in the studies by Paudel et al. (2016) in Nepal and Bhandari et al. (2022) in western Nepal. Our field observations also showed that there was a similar type of trend in Panchase and surrounding areas, where the local people left their productive land and migrated to the city. Population density increased vigorously hence increased in the settlement (86.55%) within 20 years in the city area of Byas municipality, Tanahun district as people migrated from nearby hills. Similarly, an increase in population was observed in the Kathmandu valley (412% increased) within 1989 to 2016 (Ishtiaque, Shrestha & Chhetri, 2017) and in the Pokhara valley (125.55% increased) from 1990 to 2013.

The classified images of Nepal clearly showed 48.6% of the forest area lost from 1930 to 2014 (Reddy et al., 2018). But this loss was very low from 2005 to 2014 (only 4 km2 per year). From 2005 onwards the deforestation rate is decreased due to the effective implementation of community forestry program by the government of Nepal (Nepal Ministry of Forests and Social Conservation (MoFSC), 2016). The forest loss during recent years is due to developmental projects and are comparatively more in the Terai region (Reddy et al., 2018). However, the land use/land cover change analysis of the CHAL area (landscape includes 19 districts from Terai to high mountains) between 1990–2010 showed an increased in forest area by 0.3% while the grasslands decreased slightly (WWF, 2013a). The overall forest of mid-hills of CHAL area is increasing while cropland and grasslands are decreasing. Land cover analysis in 2015 found that 48% of the mid-hills, 62.6% of high mountain and 6.1% of the high Himalayan area were covered by forest (Nepal Ministry of Forests and Social Conservation (MoFSC), 2015). However, the forest area of the mid-hills and high mountains were increasing while the croplands were decreasing (Nepal Ministry of Forests and Social Conservation (MoFSC), 2015) similar to this study.

Conclusions

Land cover change/land use patterns determine the spatial patterns of land cover in the central part of the Chitwan Annapurna Landscape. With an increase in elevation from south to north, land cover classes in CHAL showed a change in composition of riverine forest, barren area, croplands, developed areas, mixed forest, Sal dominated forest and grasslands. The land cover change analysis of 2000, 2010 and 2020 showed the clear scenario of land cover changes, mainly in human-dominated fragmented landscape. The results of the temporal and spatial analysis of the land cover provide the baseline information for the conservation of wildlife habitats, landscape management and sustainable development of the landscape.

Supplemental Information

Supplemental Information 1 Different bands of Landsat 5 TM, Landsat 7 (ETM) and Landsat 8 (OLI) used for Band Combination.

Click here for additional data file.

Supplemental Information 2 Land cover change from 2000 to 2020 in Old Padampur, New Padampur, Byas, Panchase Protected forest area, a part of ACA.

Click here for additional data file.

Supplemental Information 3 Error matrix resulting from classifying test pixels Accuracy assessment on the basis of ground truthing points (Land cover 2000).

Click here for additional data file.

Supplemental Information 4 Error matrix resulting from classifying test pixels Accuracy assessment on the basis of ground truthing points (Land cover 2010).

Click here for additional data file.

Supplemental Information 5 Error matrix resulting from classifying test pixels Accuracy assessment on the basis of ground truthing points (Land cover 2020).

Click here for additional data file.

Supplemental Information 6 Satellite image dataset and ground truthing points for 2020.

Click here for additional data file.

We are grateful to reviewers and subject editors for valuable comments and suggestions. Our thanks also go to field assistants, members of community forests and the staff of Chitwan National Park and Annapurna Conservation Area, who supported us during data collection. Our thanks also go to all of the participants in the participatory GIS.

Additional Information and Declarations

Competing Interests

Author Contributions

Data Availability

The authors declare that they have no competing interests.

Jagan Nath Adhikari conceived and designed the experiments, performed the experiments, analyzed the data, prepared figures and/or tables, authored or reviewed drafts of the paper, and approved the final draft.

Bishnu Prasad Bhattarai conceived, designed and supervised the experiments, analyzed the data, prepared figures and/or tables, authored or reviewed drafts of the paper, and approved the final draft.

Maan Bahadur Rokaya analyzed the data, authored or reviewed drafts of the paper, and approved the final draft.

Tej Bahadur Thapa conceived, designed and supervised the experiments, analyzed the data, prepared figures and/or tables, authored or reviewed drafts of the paper, and approved the final draft.

The following information was supplied regarding data availability:

The raw data is available in the Supplemental Files.

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
