# Peer review of "Land use/land cover changes in the central part of the Chitwan Annapurna Landscape, Nepal"

_PeerJ, doi:10.7717/peerj.13435_

## Round 0.1 · original submission · Major Revisions

I have received back the assessment by one reviewer on your manuscript. I believe this is a quite straightforward and succinct paper and the comments by one reviewer are more than enough. I have also provided comments directly on the attached pdf. Please, make sure you address those in your rebuttal letter as well.

R1 mentions that you need to better justify the division of land use classes and their ground validation. Also, adjust the title to better reflect the actual region you worked on. You also have too many figures. R1 suggested you move Figs 6-16 to the suppl mat. I notices that you made separate figs for each subregion. I highly recommend you to put all the figures subregions together in a single plate, and the plots showing the change through time in another plate.

I highly suggest you follow the structured abstract that PeerJ uses. Also, include one or two sentences at the beginning of the abstract to tell the reader your motivations and knowledge gaps you're trying to address in this study.

The arrangement of paragraphs in the discussion can be improved. See also the comments by the reviewer.

Reviewer 1 ·

Basic reporting

Thank you for the opportunity to review this interesting manuscript. It is a much-needed study providing baseline information on the status of land cover and changes over time. From the title, I understood that your study covered the entire Chitwan Annapurna Landscape but looking at the map, you only covered a small portion of the CHAL. Make it clear in the title and study area section.
The manuscript needs thorough English editing for clarity.

Experimental design

The study used the standard method for land use land cover mapping. However, it is not well described the basis for the eight land cover classes and how the ground control points were collected within these land cover classes. Some of the class such as mixed forest represents different species combination in different areas (lowland and hills). How it is represented in the mapping has not been defined well. Methods section does not say anything about the subset analysis (Padampur, Byas and other areas), how they were defined and what was the logic behind those analyses. The results of the sub-sets can be presented in a table in summarized form. The figures from 7 to 16 can go in the supplementary materials.

Validity of the findings

The discussion section needs more work. Start with the highlights of the results and discuss about the possible causes and explanation of the changes in the forest cover in the past 20 years. You can also do a comparison between different sites where separate analysis was run. Some ideas appear abruptly in the conclusion section but it is not discussed in the previous sections.

Additional comments

Some minor comments are as follow

L46: LULCC can be positive or negative for forest loss. LULCC does not cause the biodiversity loss but forest loss cause, please correct.
L51-53: Too long sentence, restructure for clarity
L54-56: All are not reliable, it depends on quality of data
L86-87: There is updated eco-regions, please see Dinerstein et al. 2017 (https://academic.oup.com/bioscience/article/67/6/534/3102935), and it is not WWF only.
L110: successive is not appropriate
L113-114: Google earth is a platform, does not develop maps.
L134-136: Please provide information which layers were staked (all or some specific bands).
L142: How these points were collected? Did you look for specific number of points for the different land use classes or collected randomly/systematically?
L146: Provide details of pGIS, how many group discussions, whether conducted randomly or systematically? How you covered the study area?
L179-183: The sentence is too long and not clear what it means.
L186: The topographic maps do not differentiate the forest types, how did you used it for accuracy assessment for various forest types?
L217: Is this result for 2020 or …?
L222-225: mentioned only six land cover types, what happened with other two?
L232-260: Not mentioned anything about the separate analysis in the methods section. How and why did you conducted this analysis? I think this can be summarized in one table.

---

## Round 0.2 · accepted · Accept

Thank you for implementing all the changes suggested by the reviewer and I. I believe this revised version of the manuscript is much improved and adequate for publication.